# Formulation of Antioxidant Gummies Based on Gelatin Enriched with Citrus Fruit Peels Extract

**DOI:** 10.3390/foods13020320

**Published:** 2024-01-19

**Authors:** Francesca Aiello, Paolino Caputo, Cesare Oliviero Rossi, Donatella Restuccia, Umile Gianfranco Spizzirri

**Affiliations:** 1Department of Pharmacy, Health and Nutritional Sciences, University of Calabria, 87036 Rende, Italy; francesca.aiello@unical.it (F.A.); donatella.restuccia@unical.it (D.R.); 2Department of Chemistry and Chemical Technologies & UdR INSTM, University of Calabria, 87036 Rende, Italy; paolino.caputo@unical.it (P.C.); cesare.oliviero@unical.it (C.O.R.); 3Ionian Department of Law, Economics and Environment, University of Bari Aldo Moro, 74123 Taranto, Italy

**Keywords:** citrus, polyphenols molecules, gelatin, gummies, circular economy, agri-food waste

## Abstract

In this work, the peels of red and blonde oranges as well as lemons were efficiently (5.75–9.65% yield) extracted by hydroalcoholic solution with ultrasound assistance and employed as active molecule sources in the preparation of functional gummies. Antioxidant performances of the hydroalcoholic extracts were characterized by colorimetric assays, whereas LC–HRMS analyses identified the main bioactive compounds (phenolic acids and flavonoids). The highest scavenging activity was recorded for lemon extract in an aqueous environment (IC_50_ = 0.081 mg mL^−1^). An ecofriendly grafting procedure was performed to anchor polyphenols to gelatin chains, providing macromolecular systems characterized by thermal analysis and antioxidant properties. Scavenger abilities (IC_50_ = 0.201–0.454 mg mL^−1^) allowed the employment of the conjugates as functional ingredients in the preparation of gummies with remarkable antioxidant and rheological properties over time (14 days). These findings confirmed the possible employment of highly polluting wastes as valuable sources of bioactive compounds for functional gummies preparation.

## 1. Introduction

Citrus fruits, a prominent category in fruit and vegetable production and trade, reached a global output of 162 million tonnes in 2021 [1]. Oranges dominate, comprising over 50% of global citrus cultivation, followed by tangerines, lemons, and grapefruits [2]. China leads in citrus production, contributing 28.8% with 46.7 million tonnes in 2021 [3]. Brazil, India, Mexico, and Spain are other major producers, collectively representing 58.8% of global citrus farming [3]. Citrus fruits are usually processed for juice making, jams, essential oils, and confectionaries [4]. However, the citrus industry produces high amounts of waste, accounting for about 50% of the total fruit, mainly peels, seeds, pomace, and wastewaters [5,6,7]. In particular, peels reach 10 million MT each year, representing a heavy environmental burden related to high costs of management [8,9]. Nevertheless, citrus peels can represent a valuable raw material owing to their composition, rich in natural antioxidants, pectin, enzymes, and essential oils using advanced methods [10,11]. Efficient extraction of valuable compounds, like polyphenols, alkaloids, and essential oils, from orange and lemon peels requires optimizing parameters, such as solvent choice, temperature, time, and pressure [12,13,14,15]. The emphasis is on refining large-scale extraction with aqueous solvents for safety and economic advantages [16,17,18,19,20,21].

The antioxidant properties of the polyphenols in these extracts counter pathologies linked to excessive reactive species, especially free radicals [9,22,23]. These antioxidants neutralize free radicals, mitigating their harm by forming stable species. Green ultrasound extracts from citrus peels, abundant in polyphenols, limonoids, and polymethoxyflavones, show potential applications in various fields, including the food sector [24,25,26]. Ultrasound, with high frequency and low temperatures (32 °C), enhances solvent penetration, releases bioactive molecules, and prevents the loss of volatile compounds in orange and lemon peels [27,28,29]. 

To exploit the antioxidant properties of this class of molecules, the peels of three citrus fruits, red orange (RO), blonde orange (BO), and lemon (L), were extracted by an ultrasound-assisted procedure using an hydroalcoholic mixture. All the extracts, ROE, BOE, and LE, were chemically characterized by ultra-high-performance liquid chromatography–high-resolution mass spectrometry (UHPLC–HRMS) analysis and the antioxidant activity was evaluated by in vitro assays. 

The antioxidant extracts underwent a fully eco-sustainable molecular grafting reaction to covalently bind polyphenol molecules, through a redox couple initiated by oxygen peroxide and ascorbic acid [30], on the sidechain of gelatin, a natural polymer widely used in the food sector [31,32]. Investigating the antioxidant performance and thermal stability via differential scanning calorimetry (DSC), the polymer conjugates ROEP, BOEP, and LEP were effectively utilized as functional ingredients in crafting apple-flavored gummies, labeled ROG, BOG, and LG.

The global jellies and gummies market is projected to reach USD 6665 million by 2023, with a compound annual growth rate of 5.6% from 2023 to 2033. The rising consumer demand for fruit jelly snacks is a key driver of market growth [2]. Expansion factors include introducing new flavored varieties, developing low-calorie or vegan options, and creating functional candies with jelly-like features [33]. Incorporating vegetable extracts, especially from agricultural byproducts like waste citrus peels, strategically enhances nutritional profiles, aligning with a circular economy to minimize waste and diversify valuable product ranges [34]. Leveraging by-products for high-value gummies offers a sustainable solution, aligning with environmental and economic objectives. In this context, the goal of this research was the synthesis of a macromolecular antioxidant to be used as key ingredient in the preparation of functional gummies. The antioxidant activity of the samples, as well as their rheological properties, were investigated as a function of the storage time.

## 2. Materials and Methods

### 2.1. Materials

Gallic acid, (+)-hydrated catechin, Folin–Ciocalteu reagent, sodium carbonate (Na_2_CO_3_), sodium nitrite (NaNO_2_), sodium molybdate (Na_2_MoO_4_ּ 2 H_2_O), hydrochloric acid (HCl), sodium hydroxide (NaOH), aluminum chloride (AlCl_3_), 2,2′-diphenyl-1-picrilhydrazyl radical (DPPH), and 2,2′-azino-bis(3-ethylbenzothiazolin-6-sulfonic) radical (ABTS) were purchased from Sigma Aldrich (Sigma Chemical Co., St. Louis, MO, USA). Ethanol and purified water were purchased from VWR (Chromasolv, VWR International Srl, Milano, Italy). Apple-flavored water (from Levissima Spa, Valdisotto, Italy) was bought in a local store and used without any treatments.

### 2.2. Preparation of the Citrus Extracts

Peel samples from the waste of fresh juice making, namely of red orange (RO), blonde orange (BO), and lemon (L), ref. [24] were stored at −20 °C until extraction, without any additional treatment. The method employed was designed for these matrices. In each experiment, 20.0 g of sample, left thawed at room temperature, was added to 150 mL of 70% (*v/v*) hydroalcoholic mixture. The extraction procedure was performed by an ultrasound-assisted process using an Ultrasonic Bath ARGO^®^ at 200 W, 40 KHz, and we fixed the temperature at 25 °C for 45 min. The suspension was then filtered under vacuum using Whatman No. 3 paper filter. The extracts were concentrated by a rotary evaporator (16 mbar) at low temperatures (32–35 °C) to preserve the integrity of the components. The extracts ROE, BOE, and LE were then freeze-dried, providing a vaporous dried solid (moisture content less than 2%) stored at 4 °C until further analyses.

### 2.3. Chemical Characterization of Citrus Extracts

Citrus extracts were solubilized in methanol/water (50:50 *v/v*) at a concentration of 10 mg mL^−1^. The samples were filtered through a 0.45 μm membrane filter and analyzed by liquid chromatography–mass spectrometry. For reverse phase ultra-high-performance liquid chromatography (RP-UHPLC) analyses, a Kinetex Biphenyl with a column with geometry 100 mm × 2.1 mm, 2.6 µm (Phenomenex, Bologna, Italy) was employed at a flow rate of 0.5 mL min^−1^. The mobile phases consisted of (A) 0.1% acetic acid in H_2_O and (B) acetonitrile plus 0.1% acetic acid. Analysis was performed in a gradient as follows: 0–10.0 min, 5–30%B; 10.01–12.0 min, 30–70%B; 12.01–13.0 min, 70–98%B; 98%B hold for 1 min; returning to initial conditions in 0.1 min. The column oven was set to 40 °C, and 5 µL was injected. UHPLC–HRMS analyses were performed on a Thermo Ultimate RS 3000 coupled online to a Q-Exactive hybrid quadrupole Orbitrap mass spectrometer (Thermo Fisher Scientific, Bremen, Germany) equipped with a heated electrospray ionization probe (HESI II) operated in negative mode. The MS was calibrated by Thermo calmix (Pierce™) calibration solution. Full MS (100–1500 *m*/*z*) and data-dependent MS/MS were performed at a resolution of 35,000 and 15,000 full width at half maximum (FWHM), respectively; normalized collision energy (NCE) values of 10, 20, and 30 were used. Source parameters: sheath gas pressure, 50 arbitrary units; auxiliary gas flow, 13 arbitrary units; spray voltage, +3.5 kV, −2.8 kV; capillary temperature, 310 °C; auxiliary gas heater temperature, 300 °C. A node-based processing workflow was custom-built in Compound Discoverer^TM^ software v.3.3 (Thermo Fisher Scientific) to search and identify compounds in the citrus extracts [24].

### 2.4. Synthesis of the Antioxidant Gelatin Conjugate by Grafting Reaction

Conjugate polymers were synthesized by a method reported in the literature with some modifications [35]. In a reaction flask, 500.0 mg of gelatin was dissolved in 30 mL of purified water. Then, 12.5 mL of H_2_O_2_ 20% *v/v*) and 250.0 mg of ascorbic acid were added. The solution was kept under stirring and after 2 h, an amount of each extract (previously solubilized in 7.5 mL of purified water) equivalent to 70.0 mg of GA was introduced into the reaction flask. After 24 h, the polymer solution was purified by dialysis (MWCO: 12–14,000 Dalton) in purified water at room temperature for 72 h. The solution obtained was subsequently frozen at −18 °C and freeze-dried to obtain three vaporous solids (labeled ROEP, BOEP, and LEP). Similarly, the same procedure was performed in absence of the extract to obtain a control polymer labeled BP. 

### 2.5. Differential Calorimetric Analysis of the Functionalized Gelatins

The DSC studies were performed using a SETARAM 131 instrument. The analyzed amount of each sample was around 3–10 mg. It was enclosed in an aluminum pan in order to avoid direct contact between the sample and the furnace and/or sensor. An empty aluminum pan was used as a reference to track real heat flow of each sample. Analyses were performed by an isothermal to 25 °C for 20 min and, after, by a temperature ramp test from 25 to 650 °C at a temperature scan rate of 20 °C/min under nitrogen flux (20 sccm) [36].

### 2.6. Preparation of Gummies Based on Functionalized Gelatins

Several gummy formulations were prepared using the heating and congealing method [37]. Functionalized gelatins ROEP, BOEP, and LEP and commercial gelatin (CG) were used as gelling agents for the production of antioxidant apple-flavored gummies, following a recipe from the literature with some modifications [38]. Specifically, the recipe used 1.0 mL of commercial apple-flavored water, 0.29 g of sucrose, 50.0 mg of functionalized gelatin, and 117.0 mg of gelatin. For the preparation of the gummies, the gelatin (both functionalized and commercial) was dissolved in the apple water at 30 °C under magnetic stirring to facilitate the hydration of hydrophilic polymer. After complete dissolution, the sugar was added, and the semi-liquid mass was poured into small circular molds made of plastic with a diameter of 3.5 cm and a thickness of 0.5 cm. The samples were stored at 4 °C until analysis (0, 7, and 14 days).

### 2.7. Antioxidant Properties of Gummies

The gummies based on functionalized (labeled ROG, BOG, and LG) as well as commercial (labeled CGG) gelatins were analyzed in terms of antioxidant properties at several times (t = 0, 7, 14 days) using a procedure described in the literature with some modifications [39]. The extraction was carried out by suspending 400.0 mg of gummies in 10.0 mL of a hydroalcoholic mixture (methanol/water 80:20, *v/v*) with the addition of 0.1 mL of HCl (37% *w/w*). The mixture was sonicated for 15 min at 20 °C and kept for 24 h at 4 °C. Subsequently, the suspension was sonicated for 15 min and centrifuged for 10 min at 4000 rpm. Each extract was obtained by recovering the supernatant, which was then analyzed.

### 2.8. Total Phenolic Content

The total phenolic content (TPC) of extracts, polymers, and gummies was assessed following a method in the literature with some changes [40]. In a volumetric flask, 6.0 mL of each solution, 2.0 mL of Na_2_CO_3_ solution (2% *w/v*), and 1.0 mL of Folin–Ciocalteu reagent were combined. After two hours at room temperature, the absorbance was measured using a Jasco V-530 UV/Vis spectrometer (Jasco, Tokyo, Japan) at 720 nm against a control. This procedure was carried out in triplicate, and the TPC value was determined using a standard curve generated with gallic acid (GA) within the range of 8–40 μM (R^2^ = 0.9988). The TPC for each extract was expressed as milligrams of GA equivalent per gram of dry sample (mg GAE g^−1^). The experiments on the gummy samples were performed as a function of the time (t = 0, 7, and 14 days).

### 2.9. Phenolic Acid Content

The quantification of phenolic acid content (PAC) in the extracts and conjugate polymers was conducted using a modified Arnov test [41]. In detail, in a 10.0 mL volumetric flask were mixed 1.0 mL of the extract solution, 1.0 mL of 0.5 mol L^−1^ HCl, 1.0 mL of NaOH (4.0% *w/v*), 1.0 mL of Arnov’s reagent (composed of sodium nitrite at 0.1 mg mL^−1^ and sodium molybdate at 0.1 mg mL^−1^), and purified water. The absorbance was spectrophotometrically measured at 490 nm. The PAC value was expressed as milligrams of GA equivalent per gram of dry sample (mg GAE g^−1^), after establishing the corresponding calibration line. The experiments on the gummy samples were performed as a function of the time (t = 0, 7, and 14 days).

### 2.10. Flavonoid Content

The measurement of flavonoid content (FC) was carried out using a spectrophotometric method, with some modifications based on a previously published procedure [42]. In a 5.0 mL volumetric flask, 0.50 mL samples of each solution were combined with 0.15 mL of NaNO_2_ aqueous solution (15% *w/v*) and 2.0 mL of purified water. After 6 min, 0.15 mL of AlCl_3_ solution (10% *w/v*) was added. Subsequently (after another 6 min), 3.0 mL of NaOH (4% *w/v*) and purified water to reach a total volume of 5.0 mL were added. After 15 min in the dark, the absorbance of the solutions was measured using a spectrophotometer at 510 nm. The recorded result was expressed in milligrams of catechin (CT) equivalent per gram of dry sample (mg CTE g^−1^), after establishing the corresponding calibration line. The experiments on the gummy samples were performed as a function of the time (t = 0, 7, and 14 days).

### 2.11. Scavenging Activity against DPPH Radical

To assess the scavenging potential of extracts and conjugate polymers, different volumes of antioxidant species reacted with 2,2′-diphenyl-1-picrylhydrazyl radical (DPPH) free radicals to record their scavenging properties in an organic environment [43]. In each experiment, 12.5 mL of an aqueous solution of the active species was added to 12.5 mL of an ethanol solution of DPPH (200 μM) at 25 °C, and after 30 min, the absorbance was recorded (517 nm). The scavenging activity was expressed as percent inhibition of DPPH radicals, according to Equation (1):Inhibition (%) = (A_0_ − A_1_)/A_0_ × 100(1)
where A_0_ is the absorbance recorded in absence of active species, and A_1_ is the absorbance recorded in presence of antioxidant compounds.

### 2.12. Scavenging Activity against ABTS Radical

To assess the scavenging potential in an aqueous medium, different volumes of each sample (extract and conjugate polymer) solution were combined with a solution containing ABTS radicals (2.0 mL). This mixture was then allowed to incubate for 5 min at 37 °C. Subsequently, the absorbance was measured using a spectrophotometer at 734 nm [44]. The inhibition against radical species was estimated according to Equation (1). The experiments on the gummy samples were performed as a function of the time (t = 0, 7, and 14 days), by expressing scavenging activities in term of IC_30_ (30% inhibition concentration).

### 2.13. Rheological Characterization of Gummies

The rheological measurements on the gummies were carried out using a stress-controlled rheometer RS5000 (Rheometrics, Inc., Piscataway, NJ, USA) with a plate/plate geometry (gap 1.0 ± 0.1 mm, diameter 25 mm). Dynamic rheological tests (frequency sweep experiments) were performed at 25 °C, with 100 Pa of stress determined by stress sweep experiments to ensure that they were in a linear viscoelasticity regime [45,46].

### 2.14. Statistical Analysis

The inhibitory concentrations were calculated by nonlinear regression with the use of Prism GraphPad Prism, version 4.0 for Windows (GraphPad Software, San Diego, CA, USA). One-way analysis of variance (ANOVA) test followed by a multi-comparison Dunnett’s test was applied (*p* < 0.05).

## 3. Results and Discussion

### 3.1. Preparation and Chemical Characterization of the Citrus Extracts 

In this research, orange and lemon peels underwent an ecofriendly process based on ultrasound-assisted extraction (UAE) and using a water–ethanol mixture as a green extraction solvent, significantly reducing both extraction times and the volume of solvent. The extracts were recovered by filtration, concentrated, frozen, and lyophilized, while the exhausted matrix was properly disposed of. The final cryo-sublimation process allowed the obtaining of vaporous solids, with low moisture content, high solubility, and stability over time. The same extraction conditions were employed for red orange (RO), blonde orange (BO), and lemon (L) peels and the percentage yields of the process, expressed as grams of dry matter, were 5.75% (LE), 8.50% (ROE), and 9.65% (BOE), respectively. According to the literature data, both orange and lemon peels were freshly extracted, without undergoing any prior drying process, to avoid the loss of polyphenol molecules usually related to drying processes [47]. The presence of the water in vegetable cells both avoids the destruction of some phenols and aids in the extraction of active molecules. In dried matrices, all components (such as membranes and organelles) of the cells adhere to each other, making extraction with solvent more challenging [48].

Additionally, to improve the sustainability of the process, methanol, usually employed in the extraction of biologically active compounds from citrus peels, was replaced by ethanol. Li et al. (2006) studied the effect of ethanol concentration on the recovery of phenolic compounds and demonstrated that the maximum recovery of these compounds was achieved using an ethanol concentration between 62% and 85% (*v/v*) [48]. The same study showed the ability of water to assist in the extraction of phenolic compounds and other compounds highly representative in citrus peel. Therefore, considering the results of these studies, the choice of the extraction solvent fell on a water–ethanol mixture (70%) rather than the use of pure ethanol.

The UHPLC–HRMS analyses on the extracts were able to annotate 45 secondary metabolites belonging to different classes. In Table 1, the distinct profile of the analyzed extracts is reported.

Metabolites were annotated based on their accurate mass and MS/MS fragments by matching their spectra with those reported in the mzCloud repository. Figure 1 shows the base peak chromatograms for each extract and the corresponding labeled peaks are reported in Table 1. According to data in the literature, the presence of these molecules in the extracts of orange and lemon peels were confirmed as this natural matrix represents a valuable source of phenolic compounds [49]. 

The most representative phenolic classes include phenolic acids and flavonoids. It has been demonstrated that the quantities of these compounds vary during the fruit maturation process, decreasing as the ripeness increases. In fruit peels, phenolic compounds are found not only as free phenols but also as esterified, glycosylated, and insoluble phenols, and it seems that these conjugates exist at higher levels [49]. Different phenolic compounds were identified in orange peels, including 4-hydroxycinnamic acids (ferulic acid, *p*-coumaric acid, sinapic acid, and caffeic acid), 4-hydroxybenzoic acids (syringic acid, vanillic acid, *p*-hydroxybenzoic acid, and benzoic acid), six flavanones (esperetin, hesperidin, neohesperidin, naringenin, naringin, and diosmin), flavonols (quercetin and rutin), and flavones (roifolin, apigenin, and luteolin) [49]. The profiles and content of these phenolic compounds varied in terms of free phenols, esterified phenols, glycosylated phenols, and insoluble phenols compared to the amounts present in flavedo and albedo during different fruit development periods.

The analysis highlights the distinct profile of the extracts from red orange and blonde orange peels: although they are different cultivars belonging to the same species, BOE is significantly richer in compounds compared to the ROE extract. The qualitative difference in the two extracts is undoubtedly the basis for the different antioxidant activity recorded.

### 3.2. Antioxidant Performances of the Extracts

Total polyphenols quantification (TPC) of ROE, BOE, and LE was performed with a Folin–Ciocalteu colorimetric assay, and the recorded results are reported in Table 2 as milligrams of gallic acid equivalent per gram of extract (mg GAE g^−1^).

The data analysis highlighted that the quantity of available phenolic groups was higher in the lemon peel extract compared to the extracts from the peels of the two orange varieties. To better define the phenolic composition of the extracts, phenolic acids (PAC) and flavonoid (FC) compounds were also quantified, and the values are reported in Table 2, expressed as milligrams of gallic acid equivalent per gram of extract (mg GAE g^−1^) and milligrams of catechin equivalent per gram of extract (mg CTE g^−1^), respectively. Phenolic acids constitute a subclass of polyphenols characterized by the presence of at least one hydroxyl-substituted phenolic group and a carboxylic group. They can be commonly found in plant matrices bound to small organic acids (quinic, maleic, or tartaric) or linked to structural components of plant cells (cellulose, proteins, or lignin) and are rarely found in their free form [50,51]. The results of the Arnov assay demonstrate that the content of phenolic acids in BOE and LE is very similar and corresponds to approximately 40% of the total phenols. On the contrary, ROE returned a different result, with a content of phenolic acids corresponding to approximately 22% of the total phenols. Additionally, the determination of FC, usually responsible for the orange–red–yellow color of citrus fruits, was performed. The FC value recorded for BOE and LE was quite similar and almost 2.5 times higher with respect to the ROE value. The same order of magnitude was recorded in different studies reported in the literature concerning citrus peel extracts. In particular, *Citrus reticulata* peel ethanol and hot water extracts displayed a TPC value in the range of 4.24–6.24 mg GAE g^−1^, about four times lower with respect to the data recorded in this study, highlighting that the ultrasound-assisted methodology, as well as the employment of an hydroalcoholic mixture, deeply improved the extraction efficiency [52]. More recently, an ethanol/water mixture (80/20 *v/v*) was successfully employed and the TPC value quantified in orange peels increased to 21.33 mg GAE g^−1^ [53]. 

A linear relationship between phenolic content and antioxidant activity was also verified for all the extracts. This analysis was carried out by evaluating the scavenger activity of the extracts against DPPH and ABTS radicals, both in organic and aqueous environments. Total radical-trapping antioxidative potential values for lemon, orange, and grapefruit peels were reported in the range of 6.720–1.667 nmol mL^−1^, surpassing values for peeled fruits. Lime, lemon, sweet orange, and grapefruit peels demonstrated lower EC_50_ values in DPPH scavenging activity compared to their juice counterparts, indicating superior antioxidant potential. The ability of the extracts to neutralize the radical species was expressed in terms of IC_50_ (mg mL^−1^) and are reported in Table 2. The extract LE (0.634 mg mL^−1^) returned the highest scavenger activity in an organic environment, with IC_50_ values almost 1.4 times lower than those of ROE, with a linear correlation with the results concerning the available phenolic groups. Data in the literature confirmed these ranges of values, also highlighting the highest performances of the lemon extract compared to the orange ones [53]. 

However, the scavenging activity against the ABTS radical can more accurately estimate the antioxidant profile of food matrices (fruit, vegetables, and beverages) [54], overcoming the interferences usually associated with the evaluation of scavenger activity against the DPPH radical [55]. Additionally, the assay of inhibition of the ABTS cationic radical is more appropriate, considering its accuracy in detecting the response generated by hydrophilic compounds. The kinetics of decolorization of the ABTS radical allows one to extrapolate a concentration of antioxidant necessary to ensure a 50% decay in the initial absorbance value in the range of 0.081–0.154 mg mL^−1^, with LE giving the best results. In general, a linear and consistent correlation with the results obtained from the Folin–Ciocalteu assay can be observed; the trend of the antioxidant profile remains the same as that reported for the assay aimed at evaluating antioxidant activity in an organic environment, although the lower IC_50_ values highlighted a better antiradical efficiency of the extracts in an aqueous environment. 

Data in the literature confirmed the remarkable scavenger activity of the citrus peels [23]. To this regard, peels from *Sweet lime*, *Lima orange*, *Tahiti lime*, *Pera orange*, and *Ponkan mandarin* exhibited higher in vitro antioxidant capacity in DPPH and ferric reducing antioxidant power assays compared to pulps. Specifically, the methanolic extracts from *Ponkan mandarin* peel showed the highest antioxidant capacity (EC_50_ equal to 0.6 mg mL^−1^), while sour orange peel exhibited the lowest (EC_50_ equal to 2.1 mg mL^−1^) among different citrus species [56]. The variation in antioxidant capacity among citrus fruit peels is attributed to differences in polyphenol composition. Furthermore, ethanol peel extracts of Baladi and Novel oranges showed DPPH scavenging activities of 69% and 59%, respectively, while the methanol extracts of fresh orange peels demonstrated EC_50_ values of 2.05 and 1.99 mg mL^−1^ in DPPH and ABTS free radical scavenging methods [57]. Antioxidant activity values for extracts from peels of sour orange (Bigarade) and sweet orange varieties cultivated in Algeria varied, with the Bigarade variety exhibiting high total phenolic content (TPC) and pronounced reducing power (EC_50_ value of 0.568 mg mL^−1^), highlighting a strong correlation between antiradical activity and TPC in citrus fruit peels [58].

### 3.3. Synthesis of Functionalized Gelatins

The idea of creating polymeric conjugates with potential applications in the food field arose to confer greater chemical stability and slower degradation to low-molecular-weight compounds when they are conjugated with high-molecular-weight molecules [59]. This research aimed to produce antioxidant biopolymers by anchoring biologically active molecules from ROE, BOE, and LE onto gelatin polypeptide chains. This synthetic approach involved grafting antioxidant molecules onto preformed polymeric matrices using ecofriendly and non-drastic conditions, overcoming challenges in monomer purification and reaction optimization usually related to the other strategies [60,61]. Gelatin’s ability to undergo grafting reactions was leveraged in the synthesis of three different polymeric conjugates (labeled ROEP, BOEP, and LEP), considered promising for their antioxidant activities and potential health-improving effects. Similarly, a control polymer (labeled BP), employing the same experimental procedure but in absence of the extract, was also synthesized.

Specifically, the use of a water-soluble and biocompatible redox pair (H_2_O_2_/ascorbic acid) as an initiator allowed the anchoring of reactive species from the extract to gelatin polypeptide chains. Unlike commonly used initiators, such as peroxides and azo compounds, which require relatively high polymerization temperatures for rapid decomposition, the employed redox pair facilitates polymerization at significantly lower temperatures, minimizing the risk of phenolic compound degradation and avoiding the generation of toxic reaction products. The extracts were used in specific weight ratios with gelatin in the three reactions. Specifically, an equivalent of 0.120 g of GA per gram of commercial gelatin was used for each extract, determined based on the Folin–Ciocalteu assay’s phenolic group values. Purification of the polymeric conjugate from unreacted molecules was achieved through dialysis. The resulting polymeric conjugate solution was then lyophilized to obtain a fluffy, soft solid, subsequently characterized qualitatively and quantitatively to assess its antioxidant capacity.

### 3.4. Characterization of Functionalized Gelatins

Calorimetric analyses were performed on gelatin polymers and in Figure 2, the heat flows are depicted. 

The trends of BP, LEP, and ROEP, although they show different profiles, are characterized by some common signals. The profile of BOEP shows a huge difference in comparison of the other ones. All peaks are endothermic. In the following table, the most significant values for each sample are reported. DSC profiles allow us to obtain important information on the thermal stability, the presence of water, and overall, the crystallization temperatures (Table 3).

Measuring the enthalpy of fusion allows us to calculate the degree of crystallinity of a substance. A higher enthalpy corresponds to a greater interaction between molecules. The crystallization temperatures (around 130 °C) in the polymers show an evident increase, which indicates that gelatin has a nucleating effect, thus favoring solidification during the melting process. The results show that, with regard to the enthalpy values measured, BP, ROEP, and LEP samples returned some peaks with the highest enthalpy values (BP: 144.8 J g^−1^ and 144.0 J g^−1^; ROEP: 400.8 J g^−1^; LEP: 247.8 J g^−1^ and 170.3 J g^−1^), while in sample BOEP, a maximum value of enthalpy equal to 34.1 J g^−1^ was recorded. 

This behavior denotes a lower level of interaction between polymer molecules for the BOEP sample with respect to the other analyzed polymeric chains.

The assessment of the antioxidant profile of water-soluble gelatin conjugates ROEP, BOEP, and LEP was performed by the same tests previously described for the extract. Nevertheless, the colorimetric assays were valuable in confirming the successful grafting reaction and ensuring that the reaction conditions did not damage the biologically active compounds present in the extracts used for the polymer conjugate (Table 4). 

The control polymer BP exhibited negative outcomes in all tests, except in the assessment of polyphenolic content (5.53 mg of GAE per gram of polymer). This result was influenced by the natural phenolic-containing amino acids, such as tyrosine, present in the commercial gelatin. Data in the literature supported the presence of tyrosine residues in gelatin, influencing the positive Folin–Ciocalteu results for the commercial gelatin (12.36 mg GAE g^−1^) [62]. Folin–Ciocalteu results for conjugate polymers highlighted a significant increase in polyphenol content compared to BP, supporting the successful binding of bioactive molecules from the extract to the polymeric matrix during the grafting reaction. The kinetic assessment of ABTS radical decolorization highlighted the superior performance of the polymeric conjugate from lemon peel extract, while ROEP and BOEP exhibited reduced antioxidant activity. In this case, the discrepancy between the scavenging activity in organic and aqueous environments appears quite evident and the antioxidant capacity of the conjugate in water is almost one order of magnitude higher than the organic one. On the contrary, control polymers BP and CG did not display any scavenging activity. The scavenging activity of the conjugates returned IC_50_ values one order of magnitude higher compared to a gelatin-grafted polymer from carob leaves extract (0.0212 mg mL^−1^) synthetized by the same procedure [30]. In this case, the composition of the extract and the ability of the compounds to undergo the radical reaction play a key role in the formation of the covalent bond between the polypeptide chain and the phenolic moieties.

### 3.5. Preparation and Characterization of Functionalized Gummies

The matching of the gelling properties of gelatin with the antioxidant properties of citrus peel extracts through grafting reactions allowed the synthesis of a functionalized gelatin useful as a macromolecular system in the preparation of high-value functional foods, significantly increasing the potential use of agri-food by-products. To this regard, approximately 50% of the candy market value is represented by gummies, highly appreciated by consumers for their texture and chewiness and having gelatin as their basic element [63]. Several studies in the literature suggest the possibility of creating gummies enriched with extracts from leaves, fruits, and plant components [64,65]. However, these studies recommend using low amounts of raw extract in their preparation, as adding high amounts tends to result in the loss of elasticity, toughness, color, and overall palatability of the food [66]. Often, the antioxidant features of the extracts do not allow the employment of reduced amounts of extract in the preparation steps. To overcome this limit, active molecules of the extract were introduced by a basic ingredient of the food product.

To emphasize the effect and benefits derived from the use of functionalized gelatins in the production of gummies, we replaced the common fruit juice or syrup with apple-flavored water. Indeed, this not only ensured the solubilization of solid components but also imparted a pleasant flavor to the food while simultaneously proving to be devoid of additional antioxidant activity. The lack of antioxidant activity in apple-flavored water was confirmed by the Folin–Ciocalteu assay.

Four gummies (ROG, BOG, LG, and CGG) were prepared and analyzed using ROEP, BOEP, LEP, and CG as gelatinizing agents, respectively (Figure 3). 

To evaluate the beneficial properties of the gummies on human health, antioxidant activity was investigated. Specifically, a Folin–Ciocalteu assay and the ABTS radical inhibition test were performed on the gummy extracts at several times (0, 7, 14 days) to also evaluate the maintenance of this property over time (Table 5). 

The Folin–Ciocalteu assay confirmed that the employment of the polymer conjugates as gelatinizing agents significantly increased the quantity of available phenolic groups in the gummies at time zero. Additionally, the gummies displayed a remarkable scavenging activity in an aqueous environment against the ABTS cationic radical (expressed as IC_30_ in mg mL^−1^). Furthermore, no antioxidant activity was recorded in the CGG sample. 

For all samples, the trend of the parameters was evaluated over a period of 14 days. The data analysis highlighted, for all samples, reductions in the antioxidant properties of the gummies as well as in the concentration of total phenolic compounds. However, LG displayed more than four times higher parameters compared to the control gummy after two weeks. Additionally, relevant antioxidant activity was also recorded in ROG and BOG, in the same trend observed for the extracts and polymeric conjugates. 

### 3.6. Rheological Properties of the Functionalized Gummies

Rheological measurements were obtained by imposing a sinusoidal stress on the samples. The resulting strain component in phase with the stress defines the storage or elastic modulus, G′, while the strain component out of phase with the stress defines the loss or viscous modulus, G″. The applied stress is within the viscoelastic region (100 Pa). All the samples were tested by stress sweep tests at 1 Hz to determine the linear sweep strain range (see, for instance, the figure). G′ and G″ were measured in the frequency range of 0.1–60 Hz (frequency sweep tests or mechanical spectra), as reported in Figure 4.

The frequency sweep tests were performed at 20, 25, 30, and 35 °C at three different storage times (0, 7, and 14 days) and the results are reported in Figure 5.

Several types of information can be extracted from rheological results. We now consider a structural analysis of jelly by comparing dynamic behaviors occurring at different temperatures and different maturation times. The viscoelastic data are interpreted in accordance with the theory of Bohlin [67] and Winter [68], which is reported in the literature as the “weak-gel model” [69]. This model provides a direct link between the microstructure of the material and its rheological properties (Table 6). 

The introduced parameters are the coordination number, z, which is the number of flow units interacting with each other to give the observed flow response, and A, a proper constant, which can be interpreted as the interaction strength between the flow rheological units.

The rheological spectra show strong changes in the gel behavior according to the kind of polymers present in the formulation. Similar rheological characterization was performed in previous works [70,71]. In fact, all samples are characterized by strong gel trends. G′ is greater than G″ and almost flat in frequency [72,73]. The gummy prepared by commercial gelatin displayed a dominant elastic modulus at all temperatures except during the initial maturation process, returning the highest elastic modulus at time equal to zero days. The rheological behavior of the lemon gummy (LG) seems to have a weaker G′. In fact, the elastic modulus is the lowest one for all samples in all analyzed experimental conditions. The lemon gummy is liquid at temperatures over 30 °C and the elastic modulus is not detectable anymore. Higher G′ values can be considered indicative of a greater structuring of the candies, in terms of the greater interaction strength between the links of the formed network.

By quantitative analysis, A and z values are calculated. A and z are higher for gummies prepared by commercial gelatin. In fact, A is within a 2.4·10^6^–1.0·10^6^ range after two weeks, while z is around 100. The ROG sample shows comparable values. It is worthy to note that the BOG sample is less structured at zero days. In fact, its z value is about three, evidencing its low structural coordination. 

## 4. Conclusions

Every year, the citrus industry generates huge amounts of wastes, representing an environmental and economic problem. Expanding the use of these by-products could be the foundation for increasing their economic value. The extraction of the bioactive compounds from orange and lemon peels, their involvement in the synthesis of a macromolecular bioconjugate, and the preparation of functional gummies can represent a sustainable alternative for the exploitation of these by-products in a circular economy perspective. An ecofriendly procedure provided extracts rich in phenolic acid and flavonoid molecules. In particular, lemon peel extracts returned the best results both in terms of phenolic content and scavenging activity in aqueous and organic environments. The same trend was observed in the functionalized gelatins that were employed to prepare functional gummies. Antioxidant and rheological properties were retained over two weeks of storage time. Future studies on functional gummies will be necessary to evaluate their in vivo bioactive properties as well as consumers’ acceptance based on sensory features. 

## Figures and Tables

**Figure 1 foods-13-00320-f001:**
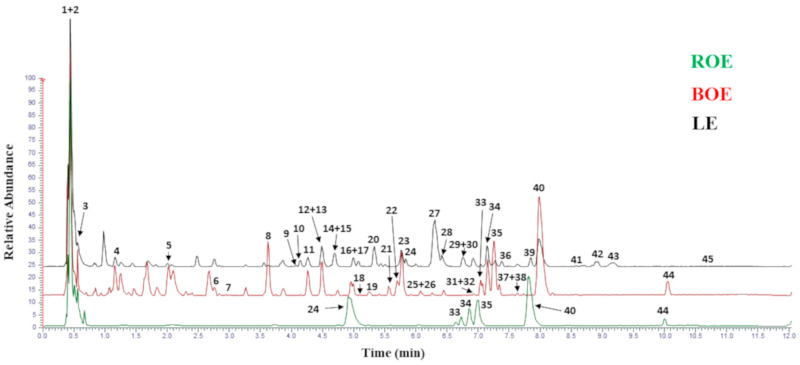
UHPLC–HRMS base peak chromatograms of ROE, BOE, and LE extracts.

**Figure 2 foods-13-00320-f002:**
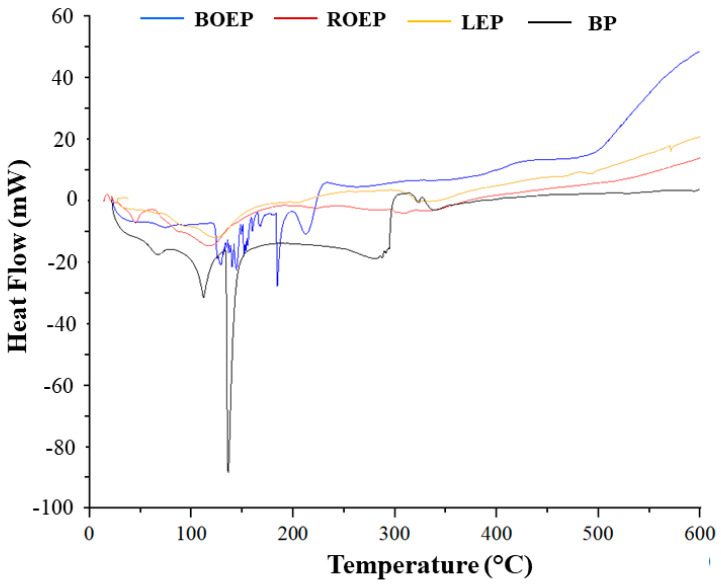
Calorimetric analyses of ROEP, BOEP, LEP, and BP.

**Figure 3 foods-13-00320-f003:**
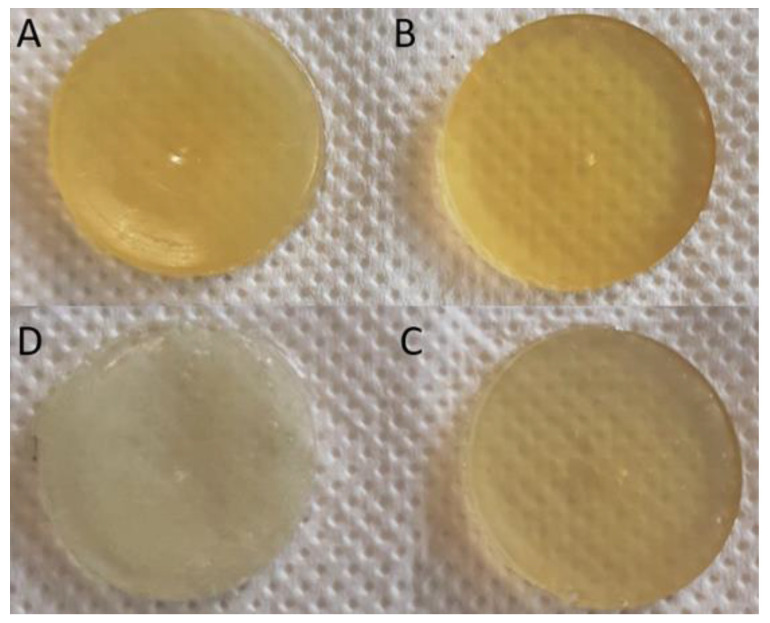
Gummies ROG (**A**); BOG (**B**); LG (**C**); CGG (**D**).

**Figure 4 foods-13-00320-f004:**
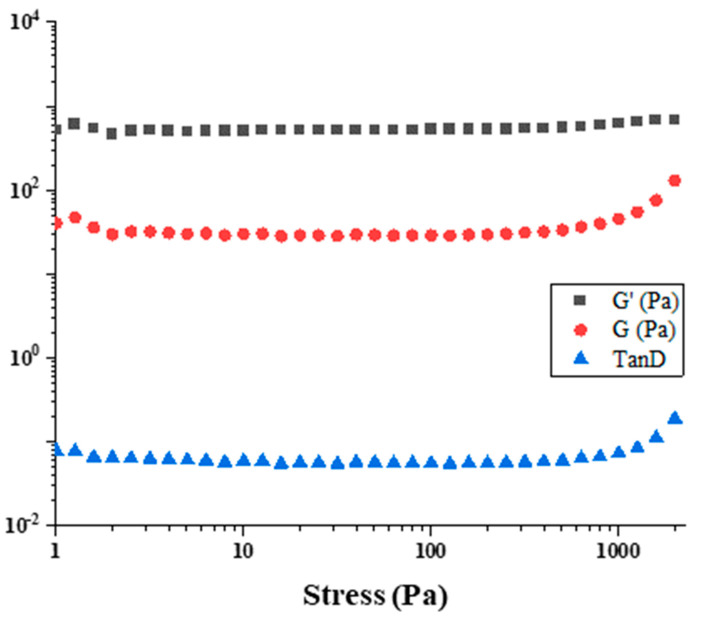
Stress sweep of the sample BOG at 25 °C.

**Figure 5 foods-13-00320-f005:**
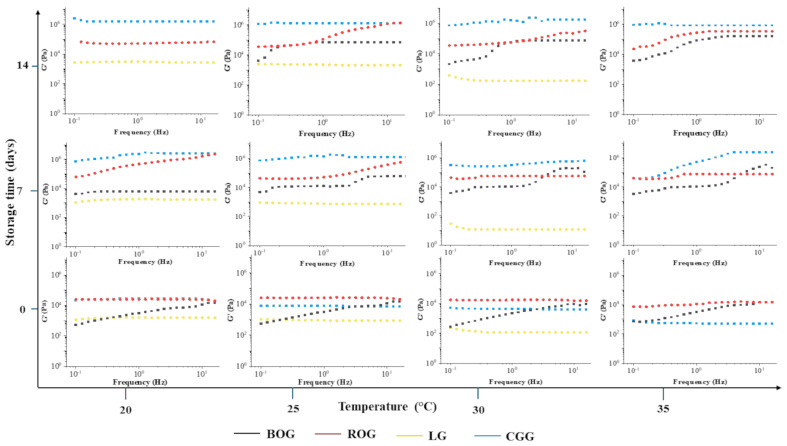
Stress sweep test of BOG, ROG, LG, and CGG at different temperatures (20, 25, 30, 35 °C) and storage times (1, 7, 14 days).

**Table 1 foods-13-00320-t001:** UHPLC–HRMS annotation of the compounds detected in extracts ROE, BOE, and LE.

Tag	Nome	*m/z*	Formula	RT (min)	Annot. Delta Mass (ppm)	ROE	BOE	LE
**1**	Gluconic acid	195.05013	C_6_H_12_O_7_	0.394	−4.59	Yes	Yes	Yes
**2**	Citric acid	191.01894	C_6_H_8_O_7_	0.441	−4.11	Yes	Yes	Yes
**3**	Gallic acid	169.01344	C_7_H_6_O_5_	0.711	−4.73	Yes	Yes	No
**4**	D-Saccharic acid	209.02982	C_6_H_10_O_8_	1.084	−2.24	Yes	Yes	No
**5**	Geniposidic acid	373.11401	C_16_H_22_O_10_	2.023	−0.02	Yes	Yes	No
**6**	Caffeic acid	179.03426	C_9_H_8_O_4_	2.789	−4.03	No	Yes	No
**7**	Fraxin	369.08279	C_16_H_18_O_10_	3.051	0.21	No	Yes	No
**8**	Quercetin 3-(2G-glucosylrutinoside)	771.19916	C_33_H_40_O_21_	3.634	0.29	Yes	Yes	No
**9**	Caffeic acid	179.03429	C_9_H_8_O_4_	3.736	−1.60	Yes	Yes	No
**10**	Tetrahydroxyflavanone 7-O-rutinoside	595.16748	C_27_H_32_O_15_	4.237	1.07	Yes	No	No
**11**	Quercetin 3-O-arabinofuranoside.	433.07748	C_20_H_18_O_11_	4.27	−0.35	Yes	Yes	No
**12**	Quercetin 3-O-arabinofuranoside.	433.07748	C_20_H_18_O_11_	4.47	−0.35	Yes	Yes	No
**13**	Vicenin 2	593.15173	C_27_H_30_O_15_	4.49	0.91	Yes	Yes	No
**14**	Quercetin-3-hexoside	463.08838	C_21_H_20_O_12_	4.684	0.39	Yes	Yes	No
**15**	Unknown	715.2457	C_32_H_44_O_18_	4.69	1.78	Yes	Yes	No
**16**	Rhamnetin 3-neohesperidoside	623.16187	C_28_H_32_O_16_	4.99	0.17	Yes	Yes	No
**17**	Kaempferol-7-O-hexoside	447.09357	C_21_H_20_O_11_	5.04	0.63	Yes	Yes	No
**18**	Quercetin 3-(2Gal-rhamnosyl-robinobioside)	755.20483	C_33_H_40_O_20_	5.151	1.08	No	Yes	No
**19**	Kaempferol-7-O-hexoside	447.09357	C_21_H_20_O_11_	5.25	0.63	No	Yes	No
**20**	Rhamnetin 3-neohesperidoside	623.16187	C_28_H_32_O_16_	5.34	0.17	Yes	No	No
**21**	Narirutin 4′-hexoside	741.2254	C_33_H_42_O_19_	5.56	0.88	No	Yes	No
**22**	Sinapinic acid	223.06079	C_11_H_12_O_5_	5.703	−1.81	No	Yes	No
**23**	Limonin hexoside	649.25451	C_32_H_42_O14	5.769	−0.28	Yes	Yes	Yes
**24**	Rutin	609.14612	C_27_H_30_O_16_	5.789	−0.18	Yes	No	No
**25**	Quercetin-3-hexoside	463.08878	C_21_H_20_O_12_	5.95	1.24	Yes	Yes	No
**26**	Quercetin-3-hexoside	463.08878	C_21_H_20_O_12_	6.034	1.24	Yes	Yes	No
**27**	Eriocitrin	595.16705	C_27_H_32_O_15_	6.31	0.35	Yes	No	No
**28**	Vicenin 2 isomer	593.15173	C_27_H_30_O_15_	6.42	0.91	Yes	Yes	No
**29**	Rhamnocitrin 3-hexoside	461.10962	C_22_H_22_O_11_	6.76	1.48	Yes	No	No
**30**	Luteolin 3′-methyl ether 7-malonylhexoside	547.1095	C_25_H_24_O_14_	6.76	0.31	Yes	No	No
**31**	Rhamnocitrin 3-hexoside	461.10962	C_22_H_22_O_11_	6.92	1.48	No	Yes	No
**32**	Luteolin 3′-methyl ether 7-malonylhexoside	547.1095	C_25_H_24_O_14_	6.92	0.31	No	Yes	No
**33**	Deacetyl nomilin hexoside	651.266	C_32_H_44_O_14_	7.05	1.92	No	Yes	Yes
**34**	Nomilinic acid hexoside	711.287	C_34_H_48_O_16_	7.15	1.67	Yes	Yes	Yes
**35**	Unknown Limonoid hexoside	843.2223	C_36_H_44_O_23_	7.25	1.27	Yes	Yes	Yes
**36**	Unknown Limonoid hexoside	843.2223	C_36_H_44_O_23_	7.34	1.27	Yes	Yes	No
**37**	Rhamnetin 3-galactoside	477.10446	C_22_H_22_O_12_	7.63	1.28	No	Yes	No
**38**	Neodiosmin	607.1699	C_26_H_32_O_15_	7.68	−1.14	No	Yes	No
**39**	Hesperetin-O-hex.-O-rhamn.-O-hexoside	771.2355	C_34_H_44_O_20_	7.85	1.24	Yes	No	No
**40**	Hesperidin	609.18219	C_28_H_34_O_15_	7.983	−0.5	Yes	Yes	Yes
**41**	Obocunone hexoside	633.2556	C_32_H_42_O_13_	8.59	1.04	Yes	No	No
**42**	Hesperidin derivative	977.29511	C_45_H_54_O_24_	8.952	1.92	Yes	No	No
**43**	Homoesperetin 7-rutinoside	623.19983	C_29_H_36_O_15_	9.791	2.7	Yes	No	No
**44**	Didymin	593.18805	C_28_H_34_O_14_	10.049	0.79	No	Yes	Yes
**45**	Catechin 7-O-gallate	441.08292	C_22_H_18_O_10_	10.626	0.44	Yes	No	No

**Table 2 foods-13-00320-t002:** Total phenolic, flavonoid, phenolic acid contents, and scavenger activity of the analyzed extracts. Data represent mean ± SD (n = 3), with different letters in the same columns being significantly different (*p* < 0.05).

Code	TPC (mg GAE/g)	PAC (mg GAE/g)	FC (mg di CTE/g)	ABTS IC_50_ (mg mL^−1^)	DPPH IC_50_ (mg mL^−1^)
**ROE**	28.33 ± 1.35 ^c^	6.37 ± 0.26 ^c^	2.41 ± 0.11 ^c^	0.154 ± 0.007 ^c^	0.861 ± 0.041 ^c^
**BOE**	33.33 ± 1.44 ^b^	13.05 ± 0.52 ^b^	5.07 ± 0.20 ^b^	0.130 ± 0.005 ^b^	0.782 ± 0.032 ^b^
**LE**	39.80 ± 1.53 ^a^	15.91 ± 0.70 ^a^	7.06 ± 0.31 ^a^	0.081 ± 0.003 ^a^	0.634 ± 0.023 ^a^

ROE = red orange extract; BOE = blonde orange extract; LE = lemon extract; TPC = total phenolic content; PAC = phenolic acid content; FC = flavonoid content; ABTS = 2,20-azino-bis(3-ethylbenzothiazolin-6-sulfonic) radical; DPPH = 2,2′-diphenyl-1-picrilhydrazyl.

**Table 3 foods-13-00320-t003:** Enthalpy and temperature values of the polymers and commercial gelatin peaks.

Sample	T Center Peak (°C)	Enthalpy (J g^−1^)
**BP**	66.0	16.8
	111.8	70.9
	136.1	144.8
	287.6	144.0
	323.6	2.9
	338.1	37.0
**BOEP**	73.9	4.0
	128.1	16.0
	139.9	3.8
	144.4	6.5
	157.6	11.1
	167.5	3.9
	184.4	22.1
	213.5	34.1
**ROEP**	45.0	37.0
	117.7	400.8
	223.0	8.1
	306.0	8.5
**LEP**	64.4	0.9
	124.5	247.8
	338.5	170.3
	466.3	5.0
	494.6	4.1

ROEP = red orange extract gelatin conjugate; BOEP = blonde orange extract gelatin conjugate; LEP = lemon extract gelatin conjugate; BP = blank polymer.

**Table 4 foods-13-00320-t004:** Total phenolic, flavonoid, phenolic acid contents, and scavenger activity of functionalized and control gelatins. Data represent mean ± SD (n = 3), with different letters in the same columns being significantly different (*p* < 0.05).

Code	TPC (mg GAE/g)	PAC (mg GAE/g)	FC (mg di CTE/g)	ABTS IC_50_ (mg mL^−1^)	DPPH IC_50_ (mg mL^−1^)
**ROEP**	16.81 ± 0.71 ^c^	4.32 ± 0.16 ^c^	2.12 ± 0.08 ^c^	0.454 ± 0.018 ^c^	2.453 ± 0.117 ^c^
**BOEP**	27.19 ± 0.94 ^b^	11.82 ± 0.48 ^b^	3.21 ± 0.12 ^b^	0.310 ± 0.012 ^b^	1.468 ± 0.061 ^b^
**LEP**	31.41 ± 1.12 ^a^	14.06 ± 0.59 ^a^	6.42 ± 0.21 ^a^	0.201 ± 0.008 ^a^	1.341 ± 0.054 ^a^
**BP**	5.53 ± 0.13 ^e^	-	-	-	-
**CG**	12.36 ± 0.49 ^d^	-	-	-	-

ROEP = red orange extract gelatin conjugate; BOEP = blonde orange extract gelatin conjugate; LEP = lemon extract gelatin conjugate; BP = blank polymer; CG = commercial gelatin; TPC = total Phenolic content; PAC = phenolic acid content; FC = flavonoid content; ABTS = 2,20-azino-bis(3-ethylbenzothiazolin-6-sulfonic) radical; DPPH = 2,2′-diphenyl-1-picrilhydrazyl. (-) = not detectable or under the limit of quantification.

**Table 5 foods-13-00320-t005:** Total phenolic, flavonoid, phenolic acid contents, and scavenger activity of the prepared gummies as function of the time. Data represent mean ± SD (n = 3), with different letters in the same columns for each time being significantly different (*p* < 0.05).

Time (Days)	Code	TPC (mg GAE g^−1^)	ABTS IC_30_ (mg mL^−1^)
**0**	ROG	2.45 ± 0.09 ^c^	6.34 ± 0.24 ^a^
	BOG	3.41 ± 0.12 ^b^	5.20 ± 0.21 ^b^
	LG	5.19 ± 0.14 ^a^	1.22 ± 0.04 ^c^
	CGG	1.71 ± 0.05 ^d^	-
**7**	ROG	2.29 ± 0.05 ^c^	6.83 ± 0.28 ^a^
	BOG	2.97 ± 0.12 ^b^	5.80 ± 0.21 ^b^
	LG	4.73 ± 0.17 ^a^	1.56 ± 0.05 ^c^
	CGG	1.46 ± 0.05 ^d^	-
**14**	ROG	1.33 ± 0.04 ^c^	8.28 ± 0.34 ^a^
	BOG	2.54 ± 0.09 ^b^	7.01 ± 0.26 ^b^
	LG	4.19 ± 0.15 ^a^	2.93 ± 0.11 ^c^
	CGG	0.99 ± 0.03 ^d^	-

ROG = red orange gummy; BOG = blonde orange gummy; LG = lemon gummy; CGG = commercial gelatin gummy; TPC = total Phenolic content; ABTS = 2,20-azino-bis(3-ethylbenzothiazolin-6-sulfonic) radical. (-) = not detectable or under the limit of quantification.

**Table 6 foods-13-00320-t006:** The gel strength (A) by “weak-gel model” of Bohlin and Winter at different temperatures (20–35 °C) and times (t_0_ = 0; day; t_1_ = 7 days; t_3_ = 14 days) for all investigated samples.

A Pa × s^1/z^ (±100)	BOG	LG	ROG	CGG
**20 °C t_0_**	520	1150	25,420	21,260
**20 °C t_1_**	4040	1070	62,080	716,230
**20 °C t_2_**		2710	67,370	2,400,000
**25 °C t_0_**	520	1050	25,400	7250
**25 °C t_1_**	4730	940	44,370	722,800
**25 °C t_2_**	4070	2460	35,550	1,100,000
**30 °C t_0_**	260	230	17,600	4940
**30 °C t_1_**	3700	30	44,310	312,000
**30 °C t_2_**	2100	384	35,600	716,900
**35 °C t_0_**	700		7300	8000
**35 °C t_1_**	3200		40,950	38,300
**35 °C t_2_**	3600		23,000	890,000

ROG = red orange gummy; BOG = blonde orange gummy; LG = lemon gummy; CGG = commercial gelatin gummy.

## Data Availability

The original contributions presented in the study are included in the article, further inquiries can be directed to the corresponding author.

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
