# Peer review of "Formulation of Antioxidant Gummies Based on Gelatin Enriched with Citrus Fruit Peels Extract"

_foods, 2024, doi:10.3390/foods13020320_

Round 1

Reviewer 1 Report

Comments and Suggestions for Authors

The reviewed work focuses on the formulation of high-antioxidant gummies using gelatin enriched with citrus fruit peel extract. The manuscript is multifaceted and intriguing; however, it requires substantial editing. The M&M section, in particular, raises significant concerns.

Chapter 2.2 addresses the extraction process. Please provide additional information on the applied vacuum pressure during solvent evaporation and specify the conditions of other operations leading to the final extract.

In Chapter 2.3, there is no mention of whether the extract underwent deproteination and de-sugaring before analysis. While proteins in 70% ethanol are likely to undergo denaturation, sugars in this solution will dissolve, and their presence in the extracts has not been mentioned.

For the preparation of gummies, apple-flavored water was used, but its characteristics were not provided anywhere in the document.

Additionally, kindly expand on all acronyms upon their first mention.

Reviewer 2 Report

Comments and Suggestions for Authors

The manuscript primarily discusses the valorization of waste from various citrus fruits to produce a valuable extract with 'high' antioxidant activity, which is then incorporated into gummies. Overall, the manuscript comprehensively describes the main points of the research with a proper discussion. However, several areas need clarification:

  1. 1. Introduction: The paragraph is too long. It is suggested that the authors divide several main points in the introduction section into separate paragraphs.

  2. 2. Assessment of Antioxidant Activity: On what basis did the author use the term 'high antioxidant' for the gummies? Is there any standard for a product to be considered 'high antioxidant'? This statement should be explained clearly in the discussion section.

  3. 3. Methodology: It is suggested that the author sequence the methods starting from extract preparation, followed by sample preparation, and then continuing with analysis.

  4. 4. Line 121: Change 'Phenolic total content' to 'Total Phenolic Content.'

  5. 5. Line 154: Please separate the methodology of DPPH and ABTS into different subsections.

  6. 6. Use of IC30: Is there a specific reason why the authors used IC30 for representing antioxidant activity?
    7. Based on the IC30 data, can the gummies produced in this study be considered to have 'high' antioxidant activity?

  7. 8. Conclusion Section: Reformulate the writing in the conclusion section. The authors should summarize the significant results of the antioxidant activity of the gummies evaluated in this study.

Reviewer 3 Report

Comments and Suggestions for Authors

This manuscript aimed to prepare functional gummies with gelation and fruit peel extract. The authors brought us some interesting results, which were meaningful to the usage of fruit by-products. However, the quality of this manuscript should be improved.

1. The novelty of this manuscript should be addressed clearly.

2. The authors should provide the entire method. For instance, peels were stored at -20 °C until extraction. Thus, the thawing method should be provided. The information on the equipment used should be provided in the required format.

3. Where were the methods of 2.2 and 2.3 from? Did the authors design them, or did they come from previous references?

4. The amounts (or percentages) of compounds in Table 1 should be provided.

5. The unit should be placed in the middle of the axis in the figures.

6. The significance difference could not be carried out among the data in Table 4 at different times.

7. Fig.5 shows the stress sweep test of BOG, ROG, LG and CG at different temperatures (20, 25, 30, 35°C) and storage time (1, 7, 14 days). Nevertheless, I think it is necessary to analyze the antioxidant properties of the gummies at different temperatures and storage time, as this manuscript aimed to prepare high-antioxidant products.

Comments on the Quality of English Language

Minor editing of English language required

Reviewer 4 Report

Comments and Suggestions for Authors

In this study, extracts from citrus peels were used to make gummies, and their antioxidant potential was determined. In my opinion the quality of the manuscript in its present form needs a significant improvement.

Aims and novelty: The aim of this paper is not precisely defined; everything is stated rather broadly. Consequently, the later discussion of results is quite superficial and scattered. Also, the novelty of the paper is not emphasized. There are many recent works on the topic of utilizing by-products in nutrition. What sets this paper apart from others?

Section 2.2: It is necessary to specify the origin of the citrus peel, where it was sourced. Did the peel undergo any pretreatment before extraction, or was it extracted as a whole? What was the moisture content of the dried extract?

Line152: Where did the mention of cookies come from? Nowhere before was it stated that cookies are being made.

Section 2.9: It is necessary to specify the type of pans used, what was used as a reference pans, and what the nitrogen flow rate was.

Results and Discussion: It is necessary to focus more specifically on your results and discuss them more deeply in line with the set aim of the paper. The current state is that 50% of this section is a general overview of the literature and methods used, making it difficult for the reader to understand the essence of this paper and what the authors wanted to demonstrate.

Line 231-269, 279-287, 362-369 This text is redundant and in content is more suitable for the introduction.

Line 272: Please, list the yield for all three extracts.

Table 1. The content of detected components should also be listed in the Table.

Section 3.4 It is not clear at all what the purpose of thermal analysis is in this paper, what it specifically demonstrates and determines in the examined samples, and how it aligns with the paper's objective. This needs to be better integrated; otherwise, it seems that thermal analysis was only conducted to enhance the paper in terms of applied methods.

Line 493-505 This text is redundant and in content is more suitable for the introduction.

Table 4 - The data in the Table should undergo mixed ANOVA ("within-subjects" factor - time; "between-subjects" factor - type of sample).

Section 3.6 - Deeper discussion of the results and explanation of the impact of extracts on rheological properties are needed.

Conclusions  - When a more specific aim of the study is formulated, the conclusion will be more specific and aligned with the defined goal.

Round 2

Reviewer 1 Report

Comments and Suggestions for Authors

You can add information about the apple-flavored water producer on line 88. Good lack! 

Author Response

The authors thank the reviewer for the revision.

The producer of the apple flavored water was inserted in the revised manuscript.

Reviewer 3 Report

Comments and Suggestions for Authors

No comment.

Author Response

The authors thank the reviewer for the revision.

Reviewer 4 Report

Comments and Suggestions for Authors

The manuscript has been significantly improved and can be accepted for publication in Foods after minor revisions.

Line 92 - It is necessary to specify the origin of the samples (purchased from a local market, obtained from a factory of fruit processing, or something else).

Line 100 - It is necessary to specify the moisture content of the dried extracts.

Author Response

The authors thank the reviewer for the revision.

The required information were inserted in the revised manuscript.